# Adipose-Secreted Exosomes and Their Pathophysiologic Effects on Skeletal Muscle

**DOI:** 10.3390/ijms232012411

**Published:** 2022-10-17

**Authors:** Binglin Yue, Hui Wang, Xin Cai, Jiabo Wang, Zhixin Chai, Wei Peng, Shi Shu, Changqi Fu, Jincheng Zhong

**Affiliations:** 1Key Laboratory of Qinghai-Tibetan Plateau Animal Genetic Resource Reservation and Utilization, Sichuan Province and Ministry of Education, Southwest Minzu University, Chengdu 610225, China; 2Academy of Animal Science and Veterinary Medicine, Qinghai University, Xining 810016, China

**Keywords:** exosome, adipose tissue, skeletal muscle

## Abstract

Due to its prominent secretory activity, adipose tissue (AT) is now considered a major player in the crosstalk between organs, especially with skeletal muscle. In which, exosomes are effective carriers for the intercellular material transfer of a wide range of molecules that can influence a series of physiological and pathological processes in recipient cells. Considering their underlying roles, the regulatory mechanisms of adipose-secreted exosomes and their cellular crosstalk with skeletal muscle have received great attention in the field. In this review, we describe what is currently known of adipose-secreted exosomes, as well as their applications in skeletal muscle pathophysiology.

## 1. Introduction

In mammals, AT was once thought to be an inert storage depot for excess calories, serving for whole body energy homeostasis. Now we know that AT express and secrete diverse factors thereby directly acting on cells (autocrine) or interact with neighboring (paracrine) or distant (endocrine) cells to act a variety of physiological and pathological processes, whose unique ability make AT the largest endocrine organ in the body [1,2,3,4]. In general, AT is considered to exist in “white”, “brown”, and brown-in-white (“brite”) forms which are the signature cells of the adipose organ [5,6]. White AT is composed of distinct cell types such as mostly mature adipocytes, preadipocytes, endothelial cells, fibroblasts, and immune cells, in which, mature adipocytes in particular are major secretory cells and release a multiplicity of heterogeneous bioactive factors termed adipokines [7,8,9]. The currently known adipokines are typically classified to several groups, such as cytokines (TNFα, IL-6, or MCP-1) [10,11,12], vasoactive factors (PAI-1 or angiotensinogen) [13,14], lipoprotein metabolism regulators (CETP or LPL), and other specifically secreted hormones (leptin or adiponectin) [15]. Through the release of these adipokines, AT establishes a crosstalk with itself and other tissues or organs such as skeletal muscle, immune system, central nervous system, liver, and pancreas, where they can exert a wide range of biological actions, including metabolic homeostasis, inflammation, and insulin resistance [13]. In addition, the quality, location, structure, and cellular composition of AT can influence their secretory profile and pathophysiologic effects [16]. Followed by a parallel development with AT, skeletal muscle has emerged as an extremely active endocrine organ secreting a number of biologically active “myokines” (including cytokines, chemokines, growth factors, and hormones), that have been proposed as the key signaling and mediator molecules for skeletal muscle to establish feedback loops with other distant target organs or tissues [17,18,19]. In recent years, the concept of adipocyte-muscle crosstalk with a predominantly adipocentric view has become a considerable focus of interest, starting with studies using co-culture models of human adipocytes and myocytes, though a bi-directional communication between fat and muscle cells takes place [20,21].

Cell-to-cell communication in mammals is generally mediated by tunnelling nanotubes or gap junctions, in recent years, exosome has emerged as a new natural carrier system during cellular crosstalk [22]. Accumulating evidence suggests that cells can secret exosomes, a kind of extracellular vesicles with about 100 nm in diameter that protect carried contents from degradation and facilitate intercellular transmission in an autocrine, paracrine, and endocrine manner. Notably, exosome-carried contents such as nucleic acids, proteins, and lipids are highly heterogenous, depending on the origin and physiopathologic conditions of donor cells [23]. Considering the extremely secretory activity of AT, there is growing interest in studying the characteristic and function of adipose-secreted exosomes, as well as their roles in adipocyte-muscle interaction [24]. In this review, we introduce the origin and trafficking of adipose-secreted exosomes and their recent advances in research on skeletal muscle pathophysiology.

## 2. Composition of Exosomes

Numerous studies show that the communication between the adipocyte and other types of cells is maintained partially through the secretion of exosomes containing distinct biomolecules [25]. Given the universality and particularity of adipose-secreted exosome cargoes, it was classified into two categories. The composition of universal cargoes is largely determined by the biogenesis of exosomes [26]. Because of their endosomal origin, all exosomes contain protein families associated with the formation of ILVs (Tsg101, tetraspanins, and Alix), membrane transport and fusion (annexins, GTPases and flotillins), heat shock proteins (Hsc70, Hsp90), major histocompatibility complex (MHC) proteins (MHC I and MHC II), and cytoskeleton proteins (actin, myosin, and tubulin), which are usually regarded as markers to identify exosomes [27,28]. In addition to proteins, exosomes are enriched in lipids such as cholesterol, glycosphingolipid, phosphatidylserine, and ceramide, which are conserved and essential for maintenance of exosome morphology [29]. Notably, exosomal cargoes are normally recruited during ILVs formation, in which, membranes of early endosomes partly invaginate and reverse bud into surrounding lumina with cytoplasmic content to form ILVs, and a large pool of specific molecules in cytoplasm are recruited into exosomes by different mechanisms [30]. Recently, a variety of RNAs including mRNA and non-coding RNAs (rRNAs, tRNAs, snoRNAs, snRNAs, piRNAs, miRNAs, lncRNAs, and circRNAs) have been identified in exosomes. These RNAs are transferred from parent cells to recipient cells through exosomes and can exert special functional roles [31,32,33]. For adipose-secreted exosome, these specific cargoes are mainly molecules related to the metabolites secreted by adipocyte and lipid storage [34].

## 3. Biogenesis of Exosomes

Exosomes are formed during an endosomal process such as inward budding of the plasma membrane to successively form early endosomes, intraluminal vesicles (ILVs), and multivesicular bodies (MVBs), followed by recycling, degradation, or releasing exosomes into the extracellular space [35,36] (Figure 1). Membranes of early endosomes partly invaginate and reverse bud into surrounding lumina with cytoplasmic content to form ILVs, which gradually mature into MVBs. Immediately, MVBs containing dozens of ILVs fuse with the plasma membrane and secret exosomes [37]. As reviewed recently, exosome biogenesis and secretion require the involvement of an endosomal-sorting complex that is required for transport (ESCRT) which in vertebrates comprises around 30 proteins, assembles into four complexes (ESCRT-0/I/II/III) and associated proteins (VPS4, Tsg101 and ALIX) [38]. ESCRT components promote the budding and release of ILV into the endosome lumen, promoting the biogenesis of MVBs. Specifically, ESCRT-0 sorts ubiquitinated cargo proteins into the lipid domain, ESCRT-I and ESCRT-II induce membrane deformation to form the stable membrane neck, and recruitment of the Vps4 complex to ESCRT-III drives vesicle neck scission and the dissociation and recycling of the ESCRT-III complex [39]. In addition, a wide array of studies have revealed that a complete disruption of ESCRT function does not abolish ILV formation, suggesting an ESCRT-independent pathway in exosome biogenesis, and functional studies have implicated lipids in ESCRT-independent exosome formation [40,41]. The biophysical properties of lipids are crucial for cellular membrane remodeling, and the presence of ceramide on limiting membrane induces the spontaneous budding-in process to produce ILVs [42]. Through its cone shape, the ceramide from sphingomyelin by sphingomylinase directly promotes ILV membrane budding [43]. Moreover, the ceramide can be metabolized into sphingosine and sphingosin1-phosphate (S1P) in the presence of ceramidase and sphingosine kinase, which binds to S1P-receptors to the limiting membrane to further promote exosomal biogenesis [44]. Mature MVBs have different fates that, to be delivered to lysosomes or autophagosomes for degradation, fuse with the plasma membrane to release the contained ILVs as exosomes, or are transported to the trans-Golgi network (TGN) for endosome recycling [38,45]. The final secretion of exosomes depends on transport and plasma membrane fusion of these secretory MVBs, which is mediated by several factors including cytoskeleton, molecular motors, small GTPase, Rab family, and SNARE protein family, as described in detail in our previous review [46]. Eventually, signals from exosomes are transported from the donor cell to the receiving one through three different mechanisms: endocytosis, direct membrane fusion, or receptor-ligand interaction, thus influencing major biological processes [47,48,49].

## 4. Adipose-Derived Exosomes in Adipocyte–Myocyte Crosstalk

Knowledge about developmental biology suggests the growth of skeletal muscle preceding that of AT; however, with age the growth speed of AT increases, while that of skeletal muscle decreases [50]. The relative proportion of these different tissues is regulated throughout life, and changes in their proportion at different developmental stages may affect relevant metabolic regulation, contributing the occurrence of disease. The development of AT and skeletal muscle is interdependent and continuous, and muscle-fat crosstalk has been a hot topic for the last ten years. According to previous research, an extensive crosstalk between AT and skeletal muscle was supported by muscle and fat co-culture systems. Co-culture systems are normally designed to analyze the interactions between two different cell types, and the understanding of crosstalk between C2C12 and 3T3-L1 cells can be useful in addressing several questions related to muscle modeling, muscle degeneration, apoptosis, and muscle regeneration [51]. AT is a loose connective tissue composed of many secretory cells, besides mature adipocytes, there are also non-adipocyte compartments including preadipocytes, AT-derived stem cells (ADSCs), AT macrophages (ATMs), and AT endotheliocytes (ATEs) [52]. The heterogeneity of cells in AT makes the crosstalk between AT and skeletal muscle more complicated.

In 2009, adipose-derived exosomes were first found, which can act as a mode of communication between AT and macrophages [53]. After that, accumulating evidence indicated that adipose-derived exosomes can transfer specific cargoes from cell to cell via paracrine or endocrine signaling, thereby regulating metabolic inflammation, energy metabolism, and insulin sensitivity of skeletal muscle [24,54]. At present, the study of AT-derived exosomes on skeletal muscle mainly focuses on the research of the cargoes (proteins and miRNAs) contained in the exosomes. Here, we review the existing evidence about exosomes derived from different cellular origins in AT and their roles in regulating the pathophysiology of skeletal muscle (Table 1, Figure 2).

## 5. Proteins

In addition to conventional secretion pathways via the ER, TGN, carrier proteins, or lysosomal compartments bypassing Golgi processing, cytosolic proteins can be shuttled between cells through exosomes [55,64]. The proteins were divided into two parts in consider of the universality and particularity of exosomal proteins in AT. The composition of universal proteins is mainly caused by the biogenesis of exosomes, including some membrane proteins and key proteins of exosomes formation. For instance, adipocyte-secreted proteins such as adiponectin, resistin, perilipin A, and fatty acid binding protein 4 (FABP4) have been identified as the markers of adipocyte-derived exosomes [65]. A recent study verified that ADSC-derived exosomes are enriched with the exosomal markers TGS101, CD9, CD63, HSP90, and ALIX [66]. Moreover, AT endothelial cells were identified as a source of adipose-derived exosomes, which are positive for the markers CD9, CD63, TSG101, and ALIX [67]. Among these exosomal proteins, adiponectin and FABP4 are highly enriched in adipocytes, serving as a specific marker for adipose exosomes.

The specificity of exosomal proteins in AT is reflected in their different sources of species and cells, determining the differences in their biological functions. Alterations in the mass of AT result from changes in adipocyte number and size, which is known as hyperplasia and hypertrophy, respectively [68]. Adipocyte hyperplasia involves the proliferation and differentiation process from preadipocytes to mature adipocytes, and the synthesis or uptake of fatty acids leads to the accumulation of lipids in existing adipocytes, resulting in hypertrophy [69]. Hartwig et al., purified exosomes from cultivated primary human preadipocytes, and characterized their proteomic profiling in comparison with the overall secretome of the same cells by LC-MS. A total of 884 proteins were identified in their analysis, in which, 212 proteins are commonly found in both AT and released exosomes are mainly involved in inflammatory and fibrotic processes, whereas 672 exosome-specific proteins are rather assigned to membrane-mediated processes and signaling pathways. In addition, these preadipocyte-derived exosomes are enriched in proteins lacking signal peptides that direct proteins to the traditional secretory pathway, suggesting their significant role in the overall human adipokinome [55].

It should be noted that adipocyte hypertrophy is not a synchronous process, and preadipocytes and small adipocytes can store lipids by increasing their storage capacity when larger adipocytes are no longer able to accommodate increased lipid storage [70]. When adipocytes suffer from lipid overload and subsequent adipocyte hypertrophy, hypoxia develops in AT resulting in adipocyte dysfunction [71]. A recent study reported that hypoxic 3T3-L1 adipocyte-derived exosomes were enriched in enzymes related to de novo lipogenesis such as fatty acid synthase (FASN), glucose-6-phosphate dehydrogenase (G6PD), and acetyl-CoA carboxylase (ACC), promoting lipid accumulation in non-stressed cells. Moreover, the amount of proteins secreted from these exosomes increased by 3-4-fold under hypoxia, suggesting the secretion of hypoxic adipocyte-derived exosomes may be a way to avoid lipid overload and the associated cellular stress in AT [56].

ADSCs can be easily expanded in culture in large numbers and mediate the endogenous regeneration of bone, cartilage, and skeletal muscle [72]. In a model of cardiotoxin (CTX)-induced muscle injury, proteins secreted by ADSC were found either as free entities or within exosomes, with anti-inflammatory properties, acting in a synergistic manner to facilitate skeletal muscle regeneration. The LC-MS profiling revealed the presence of heat shock proteins (HSPs) (HSP60, HSP90, HSP105) and superoxide dismutase (SOD2), that are known to resist stressful conditions, such as heat, wound healing, or oxidative stress [57]. A parallel study demonstrated that exosomes secreted by ADSCs with multiple differentiation ability show potential in regeneration and protection of skeletal muscle. Ni et al., analyzed the protein profiles of ADSC-Exos and identified 1466 proteins that are involved in various cell functions. According to these results, ADSC-Exos contained various proteins that can facilitate skeletal muscle proliferation and regeneration by stimulating related signaling pathways such as PI3K-Akt, Jak-STAT, and Wnt pathways [58].

In the pathological context, insulin resistance is a key component in the etiology of type 2 diabetes. As the major insulin-sensitive tissues, skeletal muscle was proved to be regulated by adipose-derived exosomes with their cargo during the development of insulin resistance [24]. Proteomic characterization of adipocytic exosomes from diabetes animal models (OLETF) and controls (LETO) were firstly performed in rats. In the study, 509 proteins were identified in adipocyte-derived exosomes, including 128 upregulated (i.e., caveolin 1, lipoprotein lipase, and aquaporin 7) and 72 downregulated (i.e., adenylate kinase 2, catalase, and liver carboxylesterase) in the exosomes of diabetic rats [59]. In the context of obesity, ATM was activated into proinfammatory M1 macrophages in AT, which produces a variety of proinflammatory cytokines such as tumor necrosis factor alpha (TNFa), macrophage-colony-stimulating factor (MCSF), retinol binding protein 4 (RBP-4), etc., directly inhibiting insulin sensitivity in skeletal muscle [73,74]. In vitro studies show that treatment with obese ATM-exosomes led to glucose intolerance and insulin resistance in myocytes, while lean ATM-exosomes treatment improved systemic insulin sensitivity; however, no protein cargoes in ATM-exosomes were reported [61].

## 6. RNAs

Among these molecules contained in adipose-derived exosomes, the biological response of miRNAs has been shown to play an important role in skeletal muscle development, regeneration, homeostasis, and disease pathogenesis. There is growing evidence that AT is the major source of circulating exosomal miRNAs that are capable of targeting mRNAs in recipient cells and, hence, are associated with cell-to-cell communication [75]. This was evidenced by a significant reduction in circulating exosomal miRNAs in Dicer knockout (ADicerKO) mice and humans with genetic or HIV-related lipo-dystrophy, which is associated with the decrease in Dicer, a key enzyme determining miRNAs biogenesis [76]. miRNAs are highly conserved endogenous ncRNAs (~22 nt) that negatively regulate gene expression by targeting the 3′-UTR of target genes under the control of RNA-induced silencing complex (RISC complex) [77]. Due to their high conservation across species and extensive regulatory roles in gene expression, miRNAs have attracted the most attention among exosomal cargo biomolecules. Since exosomes are involved in different biological processes due to their varied genetic materials, the RNA omics identification of exosomal components from AT is gradually emerging. In our previous study, we systematically characterized exosomal and non-exosomal miRNA profiles during the proliferation of bovine primary white adipocytes using RNA-seq. A total of 378 and 48 miRNAs were detected in adipocytes and adipocyte-derived exosomes, respectively [78]. Our results suggest that the exosomal miRNA profiles differ from that in the parent cells, and there is an active sorting of miRNAs into exosomes. A package of miRNA into exosomes occurs via the neural sphingomyelinase 2 (nSMase2)-dependent pathway, in which, nSMase2 controlling the biosynthesis of ceramide, further induce the loading of miRNAs into exosomes [79]. Moreover, some miRNAs are specifically packaged in exosomes through proteins such as Ago2, heterogeneous nuclear ribonucleoproteins (hnRNP) family proteins, and RNA-binding protein Y-box protein I (YBX1) [80,81,82]. The functions of these few exosomal miRNAs detected were predicted associated with cell proliferation and apoptotic process; however, there is no report on the effects on surrounding cells by these exosomal miRNAs in our study.

Metabolic disorders of AT usually caused the metabolic diseases in skeletal muscle, and growing evidence suggests that miRNAs can be directly delivered to skeletal muscles by negatively regulating gene expression via adipose-derived exosomes. For example, miR-27a is highly expressed in sera of obese individuals with T2DM, which may be derived from exosomes secreted by AT. 3T3-L1 Adipocytes-derived miR-27a has been shown to induce insulin resistance in C2C12 cells by targeting PPARγ, a potent modulator of whole-body lipid metabolism and insulin sensitivity [60]. Similarly, studies in vitro have suggested that ATMs-exosomal miR-155 can reduced insulin-stimulated glucose uptake in L6 muscle cells through PPARγ [61]. Moreover, as the major sources of white AT (WAT), the gonadal WAT (gWAT) secretes excessive exosomal miR-222 to promote insulin resistance in the skeletal muscle of high fat diet (HFD)-fed obese mode mice by suppressing its direct target gene, IRS1, an important regulator of insulin signaling [62].

The classic sign of sarcopenia is loss of muscle mass, or muscle atrophy, and there are many proposed causes of sarcopenia [83]. Recently, the disruption of the relationship between AT and skeletal muscle has been recognized as potential causative factors, and perimuscular AT (PMAT) increases in parallel with muscle atrophy [84]. Itokazu et al., showed that the exosomes of PMAT of aged mice contained high levels of Let-7d-3p miRNA, which decreased the number of muscular stem/progenitor cells through targeting of HMGA2, an important transcription factor required for stem cell proliferation [63].

In addition to miRNAs, circRNAs, lncRNA, tRNA, rRNA, mRNAs, etc., contents of adipose-derived exosomes have also been characterized, including their functions in recipient cells. In a mechanism known as competing endogenous RNA (ceRNA), with abundant binding sites for miRNA, circRNAs or lncRNAs can competitively bind with miRNA to repress mRNA target expression [85]. For example, exosomal circRNAs secreted by adipocytes promote the growth of hepatocellular carcinoma by targeting miR-34a and activating the USP7/Cyclin A2 signaling pathway [86]; lncRNA MALAT1 is a critical component of the human ADSC exosomes and modulates expression of mRNA involved in the inflammatory response, cell proliferation and apoptosis, and cell survival pathways [87]. In our previous report of exosomal RNA enrichment, tRNAs represented over 50% of the total small RNAs in adipose-derived exosomes, and 5S rRNA were enriched with the absence of visible 18S and 28S rRNA [78]. Similar studies have shown fragmented tRNA and rRNA in exosome may represent novel RNA types with potential regulatory functions [88,89]. Furthermore, mRNA profiles of adipocyte-derived exosomes were analyzed and these exosomal mRNAs were involved in metabolic, cellular, and immune processes [90]. A wide array of studies has suggested that exosomal mRNA profiles closely correlate with those in parental cells, which may transfer intercellular signals to affect critical biological functions. For instance, some adipogenesis-related mRNAs such as adiponectin, leptin, and PPARc2 have been detected in exosomes derived from adipogenic-induced 3T3 cells, and the expression of these mRNAs increased with the extension of induction time [90]. Given the fact that exosomal mRNAs can be translated into proteins in recipient cells, exosomal mRNAs secreted from adipocytes may be transmitted to recipient cells, where they can facilitate the protein expression programs of recipient cells [91]. Interestingly, mitochondrial DNA has been discovered in exosomes released from adipocytes in vitro [92]. However, roles of these above adipose-derived exosomal RNAs on skeletal muscle remain unclear, and further investigations are needed to expand our understanding of these exosomal RNAs in adipocyte–myocyte crosstalk.

## 7. Lipids

It has been shown that lipids are essential components of exosomal membranes, and bioactive lipids such as fatty acids, glycerides, phospholipids, sphingolipids, and saccharolipids are enrichment in exosomes compared with their parent cells [29]. Exosomes appear to carry a large amount of free fatty acids in their membranes or as a protein-bound form in their lumen. In which, most are saturated fatty acids, but unsaturated species such as arachidonic acid are also present, and exosomes can modulate endogenous cell lipid metabolism by supplying several arachidonic-derived lipid mediators to either neighboring or distant cells [93]. Moreover, exosomes derived from different carcinoma cell lines have higher levels of various types of lipids. For instance, 280 individual lipid species were identified from exosomes secreted by the prostate cancer cells, and lipid profiles revealed the significant enrichment and remodeling of sphingomyelin, phosphatidylserine, glycosphingolipids, and cholesterol compared with their parent cells [94]. In addition, exosomes secreted in the colorectal cancer cell line LIM1215 are rich in sphingolipids, cholesterol, and glycerides, especially plasmalogen- and alkyl ether-containing glycerophospholipids [95]. In this context, exosomes may vectorize bioactive lipids and involve in the process of inter-cellular communication and the onset, progression, and metastatic transformation of cancer cells. So far, the lipid composition of adipose-derived exosomes is less well-characterized, but because of their high lipids content and their crosstalk between skeletal muscle, adipose appear to be an important source of exosomal lipids that modify homeostasis in skeletal muscle, which requires further investigations.

Because of expanding capacity of adipocytes in animal tissues, ectopic intermuscular lipid deposition is frequently observed within the perimysium of skeletal muscles, which become disorganized and remodeled as adipocytes mature [96]. In vitro studies have suggested that perimuscular lipid deposition accelerates aging-induced muscle atrophy factor-related proteolysis and muscle senescence, and induces progressive loss of skeletal muscle mass and function [50,97]. Moreover, the lipid deposition in skeletal muscle can also induce a shift from type II fibers to a type I phenotype, leading to impaired muscle contractility of both type I and type II fibers, resulting in a dramatic decrease in muscle strength [98]. Growing evidence supports that fatty acids such as palmitoleate (C16: 1n7) and 12,13-dihydroxy-9Z-octadecenoic acid (12, 13-diHOME) are involved in skeletal muscle homeostasis, in which, palmitoleate can improve insulin sensitivity of a skeletal muscle by increasing glucose uptake and reducing accumulation of triacylglycerol [99], while 12,13-diHOME accelerates fatty acid uptake and mitochondrial fatty acid oxidation in skeletal muscle [100]. In addition, by stimulating toll-like receptors, accumulation of free fatty acids such as diacylglycerols (DAGs) and ceramides can induce insulin resistance in skeletal muscle [101]. It is worth noting that exosome-mediated transport can regulate endogenous cellular lipid metabolism by supplying additional substrates for eicosanoid biosynthesis and eicosanoids themselves [93]. Taken together, these studies provide the possibility of exosomal lipids-mediated crosstalk between adipose tissure and skeletal muscle. Further studies are needed to elucidate the precise lipids content of adipose-derived exosomes and their impact on skeletal muscle.

## 8. Conclusions and Perspectives

The number of exosome-related studies has been increasing over the past decade and is concentrated on the field of cell pathophysiology. Exosomes can provide a natural carrier system for cell-to-cell transfer, acting to protect loaded cargoes from being eliminated by the mononuclear phagocyte system. In this review, we focus on adipose-secreted exosomes and their contribution on pathophysiological processes of skeletal muscle. Overwhelming evidence suggests that the origin and content of exosomes all contribute to the heterogeneity of exosomes. For AT, exosomes derived from different cell types (preadipocytes, adipocytes, ADSCs, ATM, etc.) contain different properties, which have different effects on skeletal muscle. In addition, the study of AT-derived exosomes on skeletal muscle mainly focuses on the research of the cargoes (proteins and miRNAs) contained in the exosomes. The heterogeneity increases the complexity of exosome functions in crosstalk between AT and skeletal muscle. Based on the latest research progress, we introduce the biogenesis process of exosomes, discussing in detail the cargo of exosomes from various cell types in AT, and their roles in regulating the pathophysiologic development of skeletal muscle.

A literature analysis shows that the dialog between AT and skeletal muscle is bi-directional. As the major endocrine tissues in mammals, skeletal muscle also communicates with mammalian different tissues and organs to maintain systemic homeostasis. In which, skeletal muscle-derived exosomes constitute a major source of proteins, RNA, and lipids reflecting its pathophysiological state, and are able to influence the function of various target cells including adipocyte, which have been already well-described in our previous reviews [46]. However, there are some limitations of exosomes-mediated adipose-muscle crosstalk studies: (1) As mentioned above, detailed cargoes of adipose-secreted exosomes need to be further supplemented, particularly compositions of ncRNAs and lipids and their biological action in a pathophysiological context. (2) Given the current techniques limitations, it is difficult to separate high-purity adipocytes from adipose tissure, and specific exosomes subtypes from other vesicle types, resulting in the heterogeneity of cargoes in exosomes. (3) Despite accumulating evidence of the pathophysiological effects of adipose-secreted exosomes on skeletal muscle, these research are mostly limited to phenotypic investigations, and the underlying mechanisms are unclear. Overall, improvements in the identification of main molecules contained in adipose-derived exosomes are required, and whose further exploration of functional mechanisms on skeletal muscle will likely improve our understanding of the molecular pathophysiology of metabolic homeostasis.

## Figures and Tables

**Figure 1 ijms-23-12411-f001:**
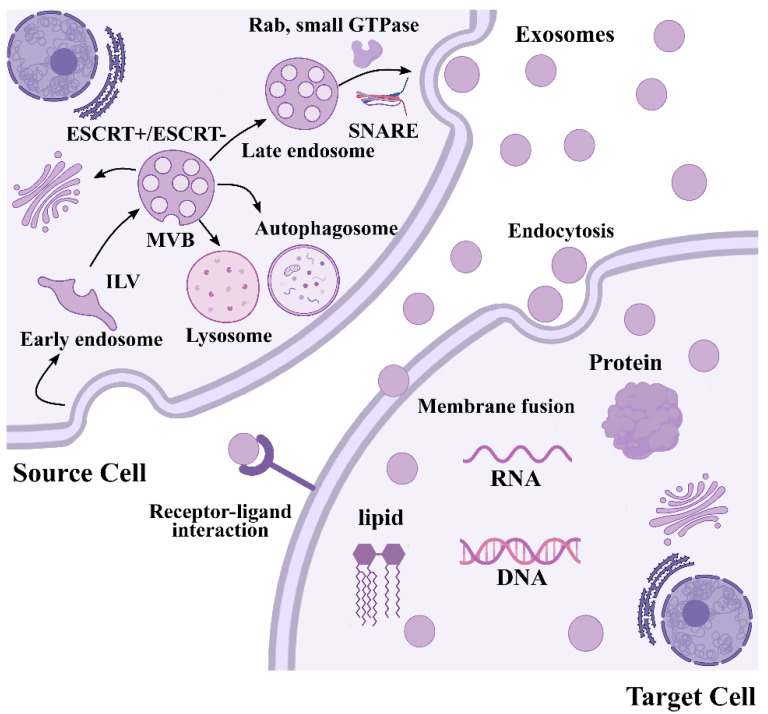
Schematic representation of biogenesis of exosomes. Exosome biogenesis starts with the invagination of the endocytic membrane and formation of ILVs inside the cell. During maturation, the cargoes (RNAs, proteins, and lipids) are incorporated into ILV through ESCRT-dependent or ESCRT-independent pathways. Then, MVBs containing exosomes are formed, followed by recycling, degradation, or releasing exosomes into the extracellular space. Finally, exosomal cargoes from exosomes are transported from the donor cell to the receiving one via endocytosis, direct membrane fusion, or receptor-ligand interaction.

**Figure 2 ijms-23-12411-f002:**
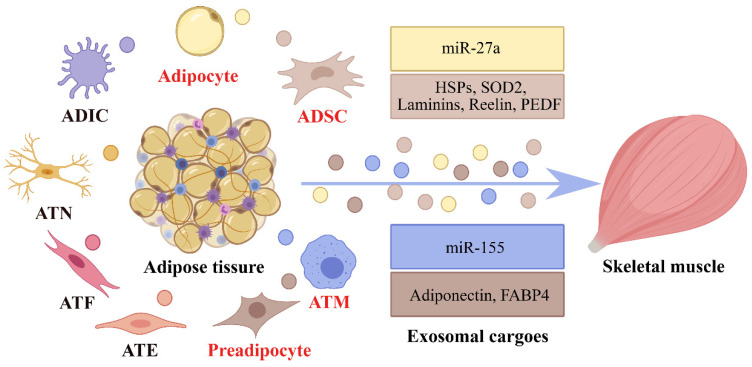
The crosstalk between adipose tissure and skeletal muscle through the adipose-derived exosomes. ADSC: AT-derived stem cell; ATM: AT macrophage; ATE: AT endotheliocyte; ATF: AT fibroblast; ATN: AT neurocyte; ADIC: AT immune cell.

**Table 1 ijms-23-12411-t001:** Summary of adipose-derived exosomes cargoes and their effects in adipocyte–myocyte crosstalk.

Cargoes	Donor	Recipient	Effects	References
Adiponectin, FABP4	Cultivated human preadipocytes	NA	Metabolic action and signaling processes	[55]
FASN, G6PD, ACC	Hypoxic 3T3-L1 adipocytes	NA	De novo lipogenesis	[56]
HSPs, SOD2	Human ADSC	Cardiotoxin (CTX)-induced skeletal muscle	Promote skeletal muscle generation	[57]
Laminins, Reelin, PEDF	Human ADSC	Skeletal muscle in rat urethra	Enhance the growth of skeletal muscle	[58]
Caveolin 1, Lipoprotein lipase, Aquaporin 7	Diabetic rat adipocytes	NA	NA	[59]
miR-27a	Palmitate-treated 3T3-L1 adipocytes	C2C12 cells	Induce insulin resistance by targeting PPARγ	[60]
miR-155	ATMs in obese mice	L6 muscle cells	Reduce insulin-stimulated glucose uptake through PPARγ	[61]
miR-222	Gonadal WAT of mice fed an HFD	Skeletal muscle of HFD-fed obese mice	Promote insulin resistance by suppressing IRS1 expression.	[62]
Let-7d-3p	PMAT of aged mice	Mouse muscular stem/progenitor cells	Inhibit cell proliferation through targeting of HMGA2	[63]

## Data Availability

Data sharing is not applicable to this article as no new data were created or analyzed in this study.

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
