# Peer review of "Adipose-Secreted Exosomes and Their Pathophysiologic Effects on Skeletal Muscle"

_ijms, 2022, doi:10.3390/ijms232012411_

Round 1
Reviewer 1 Report
In the manuscript of Binglin Yue et al. “Adipose-secreted exosomes and their pathophysiologic effects on skeletal muscle” the authors discuss the problem of the nature and mechanisms of the influence of exosomes generated by adipose tissue on skeletal muscles under normal and pathological conditions. The range of questions related to the composition of exosomes, their biogenesis, as well as the communication between the adipose tissue and skeletal muscles and the mediating role of exosomes in this process are considered. The role of proteins, RNA, and lipids of various nature in the composition of exosomes in some diseases is reviewed separately. The authors' analysis of literary sources allowed them to conclude that there is the bidirectional nature of the influence of adipose tissue and skeletal muscle. Heterogeneity of adipose tissue containing cell subpopulations having different degrees of differentiation secreting exosomes of different composition is a separate problem. In general, the problem raised by the authors has its relevance and needs further development.
The reviewer has no major remarks to the authors.
Minor remarks.
It is highly desirable to include a list of abbreviations for a better understanding of the text.
It is strongly recommended to improve the quality of the English, since the text is replete with stylistic and grammatical inaccuracies.
Author Response
The main corrections in the paper and the responds to the reviewer’s comments are as flowing:
Reviewer 1:
Comment 1: It is highly desirable to include a list of abbreviations for a better understanding of the text.
Response 1: We are very sorry for our negligence of it, and we have supplemented this part.
Comment 2: It is strongly recommended to improve the quality of the English, since the text is replete with stylistic and grammatical inaccuracies.
Response 2: We are very sorry for our incorrect stylistic feature and grammar, and we had our paper edited by a person whose first language is English according to the editor’s suggestion.
Reviewer 2 Report
This review article is fundamentally interesting, because tissue interstitial spaces are the parent/basic organization of the living body as maintaining homeostasis. Therefore, various kinds of cells (probably remaining cells during the development) located in the various organs/tissues, of course adipose tissue. As such, the basic cellular mechanism of each tissue is important. Therefore, I would like to know/clarify the following, firstly.
1. My particular concern is in the line 27-30 (Introduction). The authors saying that “mature adipocytes in particular are major secretory cells and release a multiplicity of heterogeneous bioactive factors that termed adipokines”. However, basically, mature adipocyte contains wholly lipid and have very minor cytoplasm (as represented in the Figure 2 upper light spot). This characteristic means that it is unsuited for work as secretory cells in comparison with myofibroblasts which have rich rough-ER with a large cytoplasm. Furthermore, it would be formed “exosomes” can be difficult.
2. The authors should be clarified this issue in this review, although several references were cited in the text. I think this is a big issue, because most reports have used “adipose tissue, which has contained various cells (line 27-30). Thus, I wonder if endothelial cells, fibroblasts and/or interstitial stem cells are a primary contributor to secretion.
3. In the above regard, the mechanism of represented in Figure 1 can be applied to the most of interstitial cells without adipocytes. Thus, the authors should insert “the case of adipocyte” as Figure 2, then followed by present Figure 2 as figure 3.
4. In addition, I also wonder that the adipose tissue derived stem cells (ADSC) may be wrong description, and there are stem cells located in the interstitial spaces (thus, interstitial stem cells), such as interstitial stem cells in the skeletal muscle. The latter data has been accumulated by longer terms before the ADSCs.
5. It is likely that the above regards of descriptions are slight in this article, and this affected the unbalanced references.
6. Therefore, I want to that it should be addressed first, without this step, the story wouldn't begin.
Author Response
The main corrections in the paper and the responds to the reviewer’s comments are as flowing:
Reviewer 2:
Comment 1-2: My particular concern is in the line 27-30 (Introduction). The authors saying that “mature adipocytes in particular are major secretory cells and release a multiplicity of heterogeneous bioactive factors that termed adipokines”. However, basically, mature adipocyte contains wholly lipid and have very minor cytoplasm (as represented in the Figure 2 upper light spot). This characteristic means that it is unsuited for work as secretory cells in comparison with myofibroblasts which have rich rough-ER with a large cytoplasm. Furthermore, it would be formed “exosomes” can be difficult.
The authors should be clarified this issue in this review, although several references were cited in the text. I think this is a big issue, because most reports have used “adipose tissue, which has contained various cells (line 27-30). Thus, I wonder if endothelial cells, fibroblasts and/or interstitial stem cells are a primary contributor to secretion.
Response 1-2: The research shows that mature adipocytes can secret different populations of extracellular vesicles, and different biological stimuli related to the obesity-associated chronic low-grade inflammation enhance the secretion of exosomes from mature adipocytes[1]. Mature adipocytes constitute a terminally differentiated cell population, and represent the main source of adipokines such as RBP4, Leptin, SFRP1, and Chemerin[2]. Although mature adipocyte contains wholly lipid and have very minor cytoplasm, exosomes are still being generated, and relevant mechanisms need to be further explored. Adipose tissue is composed of around 50% mature adipocytes as well as stromal cells such as adipose progenitor cells, preadipocytes, fibroblasts, immune cells, endothelial cells, and smooth muscle cells. Some studies have emphasized that adipose tissue communicates systemically with other organs (brain, liver, skeletal muscle) and locally with other cells (preadipocytes, endothelial cells and monocytes/macrophages) through secreted products[3].
Comment 3: In the above regard, the mechanism of represented in Figure 1 can be applied to the most of interstitial cells without adipocytes. Thus, the authors should insert “the case of adipocyte” as Figure 2, then followed by present Figure 2 as figure 3.
Response 3: It is really true as reviewer suggested that “the case of adipocyte” should be added, however, there are few reports on the detailed process of adipose-secreted generation. Hence, we represented the biogenesis of universal cell exosomes in Figure 1.
Comment 4-6: In addition, I also wonder that the adipose tissue derived stem cells (ADSC) may be wrong description, and there are stem cells located in the interstitial spaces (thus, interstitial stem cells), such as interstitial stem cells in the skeletal muscle. The latter data has been accumulated by longer terms before the ADSCs. It is likely that the above regards of descriptions are slight in this article, and this affected the unbalanced references. Therefore, I want to that it should be addressed first, without this step, the story wouldn't begin.
Response 4-6: Adipose tissue-derived stem cells (ADSCs) undergo self-renewal and are multipotent, with the potential to differentiate into numerous cell types, including adipogenic lineages, endothelial cells, osteoblasts, chondrocytes and myocytes[4].
We appreciate for the reviewers’ warm work earnestly, and hope that the correction will meet with approval. Once again, thank you very much for your comments and suggestions.
References:
[1] M Durcin, A Fleury, E Taillebois, G Hilairet, Z Krupova, C Henry, S Truchet, M Trötzmüller, H Köfeler, G Mabilleau, O Hue, R Andriantsitohaina, P Martin, Le Lay S. 2017. Characterisation of adipocyte-derived extracellular vesicle subtypes identifies distinct protein and lipid signatures for large and small extracellular vesicles. Journal of extracellular vesicles, 6(1), 1305677.
[2] T Romacho, M Elsen, D Röhrborn, J Eckel. 2014. Adipose tissue and its role in organ crosstalk. Acta physiologica (Oxford, England), 210(4), 733-753.
[3] Zhang Y., Yu M., Tian W. 2016. Physiological and pathological impact of exosomes of adipose tissue. Cell Prolif, 49(1), 3-13.
[4] Panina Y. A., Yakimov A. S., Komleva Y. K., Morgun A. V., Lopatina O. L., Malinovskaya N. A., Shuvaev A. N., Salmin V. V., Taranushenko T. E., Salmina A. B. 2018. Plasticity of Adipose Tissue-Derived Stem Cells and Regulation of Angiogenesis. Frontiers in Physiology, 91656.